# Kidney Biopsy in Pregnant Women with Glomerular Diseases: Focus on Lupus Nephritis

**DOI:** 10.3390/jcm12051834

**Published:** 2023-02-24

**Authors:** Gabriella Moroni, Marta Calatroni, Beatriz Donato, Claudio Ponticelli

**Affiliations:** 1Department of Biomedical Sciences, Humanitas University, Via Rita Levi Montalcini 4, Pieve Emanuele, 20072 Milan, Italy; 2Nephrology and Dialysis Division, IRCCS Humanitas Research Hospital, Via Manzoni 56, Rozzano, 20089 Milan, Italy; 3Nephrology Department, Hospital Beatriz Ângelo, 2674-514 Loures, Portugal; 4Independent Researcher, Via Ampere 126, 20131 Milan, Italy

**Keywords:** pregnancy, glomerular diseases, lupus nephritis, kidney biopsy, pre-eclampsia, chronic kidney disease

## Abstract

Despite significant improvements of renal and obstetrical management, pregnancies in women with glomerular diseases and with lupus nephritis continue to be associated with increased complications both for the mother and the fetus as compared to those of pregnancies in healthy women. To reduce the risk of these complications, planning pregnancy in a phase of stable remission of the underlining disease is necessary. A kidney biopsy is an important event in any phase of pregnancy. A kidney biopsy can be of help during counselling before pregnancy in cases of incomplete remission of the renal manifestations. In these situations, histological data may differentiate active lesions that require the reinforcement of therapy from chronic irreversible lesions that may increase the risk of complications. In pregnant women, a kidney biopsy can identify new-onset systemic lupus erythematous (SLE) and necrotizing or primitive glomerular diseases and distinguish them from other, more common complications. Increasing proteinuria, hypertension, and the deterioration of kidney function during pregnancy may be either due to a reactivation of the underlying disease or to pre-eclampsia. The results of the kidney biopsy suggest the need to initiate an appropriate treatment, allowing the progression of the pregnancy and the fetal viability or the anticipation of delivery. Data from the literature suggest avoiding a kidney biopsy beyond 28 weeks of gestation to minimize the risks associated with the procedure vs. the risk of preterm delivery. In case of the persistence of renal manifestations after delivery in women with a diagnosis of pre-eclampsia, a renal kidney assessment allows the final diagnosis and guides the therapy.

## 1. Introduction

Pregnancy is no longer considered to be a contraindication in women with glomerular diseases. However, the risk for maternal and fetal complications is increased, particularly in patients with renal insufficiency, severe hypertension, proteinuria, lupus nephritis, or a type of glomerular disease [1,2,3,4]. On the other hand, pregnancy can activate underlying diseases, as in the case of systemic lupus erythematosus (SLE) [5,6]. In addition, pregnancy induces important morphological and functional changes in the kidney, leading to glomerular hyperfiltration, albuminuria, and tubular abnormalities that may be confused with signs of activity of the underlying disease [7,8,9]. The differential diagnosis between the impairment of glomerular disease and pregnancy complications may be difficult in some cases, and a wrong decision may lead to severe consequences.

In this invited mini review, we discuss the role of the kidney biopsy in clarifying the diagnosis and evaluating the prognosis and management of pregnant women with glomerular diseases, with a focus on lupus nephritis.

## 2. Materials and Methods

We conducted a literature search from the 1990s to the present day in PubMed, Embase, and Medline and from a reference list of retrieved articles. During searching, we used these terms and keywords: pregnancy, glomerular diseases, chronic kidney disease, and kidney biopsy. The study quality and recommendations were assessed based on importance of the published studies.

## 3. Kidney Biopsy before Pregnancy

A kidney biopsy before planning a pregnancy may be indicated in women with an undiagnosed decline of kidney function, proteinuria, hypertension, or hematuria. An accurate histopathologic diagnosis may provide important prognostic and therapeutic information to facilitate appropriate counseling to prevent the risks of an unplanned pregnancy.

Pregnancy in women with autoimmune diseases, such as SLE, may present several issues. SLE may influence the pregnancy outcome, being associated with a higher than expected rate of pre-eclampsia, miscarriage, and/or pre-term delivery [10,11]. Pregnancy itself may have a deleterious impact on the immune system in mothers with SLE [5,6,12]. In addition, many drugs used to treat autoimmune diseases, including cyclophosphamide, methotrexate and mycophenolate mofetil, should be avoided in pregnancy because of fetal toxicity [13]. To reduce the risk of complications, a phase of stable remission of the underlining disease is necessary, and a kidney biopsy before pregnancy can provide a useful diagnostic opportunity to detect the activity and severity of this disease. For these reasons, pregnancy planning in SLE patients mandates the careful evaluation of renal function, urinalysis, and immunological parameters. Any alterations to these parameters should be investigated, and a kidney biopsy is strongly indicated even in the presence of mild proteinuria, as suggested by the EULAR/EDTA recommendations [14]. Actually, when mild proteinuria is present, the histological picture may be characterized by focal or diffuse proliferative lupus nephritis (LN), which requires aggressive therapy to reduce the risk of renal failure [15,16]. Nowadays, improvements of specific and supportive treatments for LN have allowed the achievement of renal remission in around fifty percent of patients, but in another quarter of patients, only a partial response can be achieved [17]. Although the achievement of a partial response is associated with a good long-term renal outcome, the residual proteinuria in these cases can be due to persistently active or to irreversible chronic lesions. Finally, severe tubulointerstitial or glomerular changes may be present despite normal or subnormal kidney function. In anticipation of pregnancy, in such situations, a kidney biopsy may suggest the need to postpone the pregnancy after a reinforcement of the immunosuppressive therapy to reduce the pregnancy complications. These cases are particularly troublesome since active nephritis at the time of conception is associated with a poor outcome in women with LN [18,19].

Additionally, among patients with primary glomerular disease, a quiescent disease and good kidney function are recommended before planning a pregnancy. In a retrospective study that included 360 patients with different histological forms of primary glomerulonephritis and normal renal function, the long-term end-stage kidney disease (ESKD)-free survival was not significantly different between patients who did and those who did not conceive after the diagnosis glomerulonephritis [4]. The same result was confirmed by two large systematic reviews and a meta-analysis of patients with IgA nephropathy [20,21]. These data suggest that pregnancy does not worsen the renal outcome if patients with primitive glomerular diseases have a normal renal function at conception. However, the identification of the histological type of glomerular disease is of particular importance. The clinical outcome is frequently poor and requires the use of corticosteroids in pregnant women with focal and segmental glomerular sclerosis (FSGS) or membranoproliferative glomerulonephritis [22,23,24,25], while it is usually better in women with IgA nephritis or membranous nephropathy, although the rate of pre-term delivery, a fetus that is small for its gestational age, and pre-eclampsia are more frequent than they are in healthy women [20,26,27]. In all these instances, a kidney biopsy before pregnancy can help with predicting the outcome, suggesting the ideal time of conception, and the best therapeutic strategy in order to minimize the risk of fetal toxicity and prevent the deterioration of the maternal kidney function.

## 4. Kidney Biopsy during Pregnancy

During pregnancy, there are hemodynamic changes characterized by an increase in the blood flow in the kidney with a consequent increase in the vascular and interstitial size up to 30%. The mean arterial pressure and systemic vascular resistance are lower during pregnancy, resulting in an increased cardiac output, renal plasma flow (RPF), and glomerular filtration rate (GFR). These changes may continue after delivery and resolve 1-month post-partum. Tubular function is also altered, leading to mild increases in the proteinuria and glycosuria levels [28,29] that may conceal the worsening or exacerbation of an underlying kidney disease or may reveal symptoms of LN or other glomerulonephritis. In doubtful cases, a kidney biopsy may provide a correct diagnosis.

Particularly difficult to recognize and differentiate from other complications are the rare cases of new-onset SLE, necrotizing or primitive glomerular diseases during pregnancy which usually develop in the first trimester [30,31,32,33,34]. De novo glomerular disease should be suspected when proteinuria or nephrotic syndrome appears or an acute kidney injury is associated with active urine sediment with dysmorphic hematuria and pleomorphic casts. A kidney biopsy can recognize the type of the glomerulonephritis and can indicate the possible treatment [35]. The disease should be managed by nephrologists in collaboration with obstetricians. LN, IgA nephropathy, and FSGS are the glomerular diseases that can present and are most frequently diagnoses during pregnancy [36], although rare cases of de novo small-vessel systemic vasculitis have also been detected by kidney biopsies and successfully treated [37]. Day et al. [38] described the presentation and the outcome of 20 pregnant women who developed renal disease and received a kidney biopsy within twenty weeks of pregnancy. Glomerular diseases were diagnosed in 19 patients and 9 of them needed therapeutic changes. Pre-eclampsia occurred in seven patients and the average fetal gestational age at delivery was 34 weeks. At last observation, 103 months after delivery, six patients were on ESKD, three had chronic kidney disease (CKD), and three had died. In another series of 15 kidney biopsies performed between 16 and 25 weeks of gestation in pregnant women with renal manifestations developed during pregnancy, three had a diagnosis of chronic glomerulonephritis, three had a diagnosis of mesangial proliferative glomerulonephritis, one had a diagnosis of diabetic nephrosclerosis, and eight had a diagnosis of LN. Five out of the eight LN patients had impaired renal function, and the others had nephrotic syndrome. In the kidney biopsy, five patients had diffuse crescents and three had a mesangial proliferative pattern. After the kidney biopsy, all LN patients were treated with methylprednisolone pulses. Renal response was not achieved in the patients with extracapillary proliferation, and a partial response occurred in the other patients. Within two years after pregnancy, four patients were in complete remission, one was on chronic dialysis, and three patients had died [39]. These data suggest a severe presentation and outcome of glomerulonephritis that develops during pregnancy and the need for rapid diagnosis with a kidney biopsy and an appropriate treatment.

More frequently, pregnant women with a diagnosis of LN or of primary glomerular diseases may exhibit increasing proteinuria, hypertension, and the deterioration of kidney function during pregnancy. These complications may be due either to a reactivation of the underlying disease (more frequently in patients with LN) or to pre-eclampsia. From a clinical point of view, a flare of lupus nephritis is usually associated with a rapid increase in proteinuria (proteinuric flares) and/or of serum creatinine (nephritic flares) [40]. In flares, arterial hypertension is frequent and the urine sediment is characterized by dysmorphic erythrocytes, polymorphonuclear cells, tubular cells, and hemoglobin or erythrocyte casts associated with other types of casts. Flares usually occur in the second or third trimester and can be mild or severe. The signs and symptoms of a lupus flare may mimic those of pre-eclampsia, requiring its recognition during pregnancy [41]. The treatment of severe flares is usually based on methylprednisolone pulses followed by oral prednisone and azathioprine. Cyclosporine or tacrolimus may also be used [42]. Pre-eclampsia is a multisystem disorder defined by new onset of hypertension with systolic blood pressure ≥140 mmHg and/or diastolic blood pressure ≥90 mmHg after 20 weeks of pregnancy. Pre-eclampsia is one of the most dangerous complications of pregnancy. It may lead to the death of both the mother and fetus and may have long-term cardiovascular and metabolic risks [43,44,45]. Pregnant women with LN are particularly susceptible to developing pre-eclampsia, especially if the disease is active and in the presence of proteinuria, hypertension, or antiphospholipid syndrome (APS) [46,47,48]. Low-dose aspirin and an early treatment of hypertension may attenuate the severity of pre-eclampsia, but this disorder remains a major cause of maternal and fetal morbidity. Intractable hypertension, seizures, thrombocytopenia, and coagulation abnormalities are life-threatening complications. Currently, there is no cure for pre-eclampsia. The prevention and management of seizures include anti-hypertensive drugs—labetalol, hydralazine, methyldopa, and/or calcium channel blockers—and an infusion of magnesium sulfate. The latter option is the ideal drug for managing pre-eclampsia, but it may be rarely cause severe cardiovascular or respiratory adverse events. Some investigators use magnesium sulfate at a dose of 1 g/hour for maintenance to prevent side effects [49]. A simple therapeutic measure is bed rest, but delivery remains the best treatment for severe eclampsia. Differentiation on clinical grounds between a lupus flare and pre-eclampsia can be a true dilemma even because lupus nephritis itself predisposes to pre-eclampsia [50]. On the other hand, this differentiation is critical since a lupus flare requires vigorous immunosuppressive therapy, whereas the delivery of the child and the placenta is the only treatment for severe pre-eclampsia. Clinically, both conditions can present with hypertension, proteinuria, and gravitational edema. The urine sediment is more frequently active in lupus flares than it is in pre-eclampsia. Thrombocytopenia is common to both of them, but microangiopathic hemolytic anemia and elevations in liver function tests are more suggestive of pre-eclampsia [42]. Lupus flares may be associated with hypocomplementemia and an increase in anti-DNA antibodies, but these parameters do show a significant association with renal disease activity [51]. The angiogenetic markers have a good diagnostic accuracy to diagnose pre-eclampsia, but they are not sensitive enough to serve as an early screening test [52]. A kidney biopsy can help with establishing a correct diagnosis and treatment in difficult cases. Flares of lupus nephritis are associated with diffuse or focal proliferative nephritis [39] associated with signs of activity, such as interstitial inflammation, neutrophil infiltration, hyaline deposits, and/or fibrocellular crescents. In pre-eclampsia, a renal biopsy is characterized by glomerular endotheliosis, the glomerulus is diffusely enlarged and bloodless, not due to proliferation, but to hypertrophy of the intracapillary cells [53]. In detail, there are swollen mesangial cells, vacuolated endothelial cells, a loss of endothelial fenestrations, and glomerular capillary occlusion. Double contour of the basement membrane in light microscopy and electron microscopy in the absence of cellular proliferation, both intracapillary and/or extracapillary, are typically associated with pre-eclampsia. Electron microscopy in pre-eclampsia is also characterized by glomerular fibrin deposits due to a non-immunologic insudation [54]. The absence of cellular proliferation in light microscopy and the absence of immune deposits in electron microscopy are crucial to differentiating pre-eclamptic disease from immunocomplex glomerulonephritis, such as LN and IgA. Electron microscopy is also useful to distinguish pre-eclampsia from FSGS, in which there are no immune deposits, but there is extensive foot process effacement.

Old and recent reports outlined fair maternal and fetal outcomes in pregnant women with glomerular diseases when hypertension was controlled and GFR was stable [55,56,57,58,59]. However, patients with nephrotic syndrome have an increased risk of severe maternal and fetal complications [60,61]. Acute kidney injury (AKI) is a rare, but serious complication of pregnancy, which causes a high rate of maternal and fetal morbidity and mortality [62]. Pregnancy-related AKI may be triggered by several causes, including hypovolemia, placental abruption, septic abortion, interstitial nephritis, pyelonephritis, obstruction of the urinary tract, or microangiopathies [63]. Clinical presentation and echography can allow a rapid diagnosis and allow the correct management of AKI in several cases, but differential diagnosis with pre-eclampsia may be difficult, particularly if the pregnant woman was affected by glomerular diseases or hypertension. A kidney biopsy may be required in these cases, as well as in women with prolonged oligo-anuria, to differentiate between tubular necrosis and cortical necrosis. Another diagnostic conundrum is represented by the differential diagnosis between pre-eclampsia and rare complications of pregnancy such as HELLP (Hemolysis, Elevated Liver function tests, and Low Platelets) syndrome, thrombotic thrombocytopenic purpura, and atypical hemolytic-uremic syndrome. These disorders may present with overlapping features, and establishing a differential diagnosis using clinical criteria can be very difficult. A kidney biopsy may be considered for the exact diagnosis and to facilitate the appropriate treatment [64], but the hemorrhagic risks of the biopsy are increased in case of severe thrombocytopenia and should be carefully evaluated [65].

## 5. Post-Partum Kidney Biopsy

Many women who develop de novo proteinuria, exacerbation of the urinary findings, or hemolytic uremic syndrome do not receive a kidney biopsy during pregnancy either because of the fear of hemorrhagic complications or because of formal contraindications related to severe hypertension or coagulation disorders. If the clinical manifestations persist after delivery, a kidney biopsy should be conducted within few months. By definition, the manifestations of pre-eclampsia completely reverse within 12 weeks after delivery. The persistence of proteinuria or of hypertension after this period require careful evaluation [41]. Several studies showed that pre-eclampsia may significantly increase the risk of chronic kidney disease and end-stage kidney disease [66,67,68,69,70]. A post-partum kidney biopsy can assess whether the previous disorders will result in recovery or tend to progression, indicating possible treatments to halt a deleterious evolution. On the other hand, patients with LN nephritis may experience an increased incidence of renal flares within the 3 months after delivery, which may require a kidney biopsy for correct diagnosis and management [71,72,73]. In our prospective study reporting the maternal outcome in pregnant women with lupus nephritis two out of fourteen renal flares (14%) occurred during the post-partum period [74]. In summary, patients that suffered a kidney injury during pregnancy, and particularly, those with persistent or increasing proteinuria after delivery should receive a regular renal check-up, including a kidney biopsy, to improve the long-term health outcomes for women with known or new glomerular diseases [75,76,77].

## 6. Complications of Kidney Biopsy

Pregnancy is regarded by many clinicians as a relative contraindication for a kidney biopsy because of the underlying risk of major complications, such as perirenal hematomas requiring blood transfusions, perirenal abscess, and even sepsis [38,78]. However, the incidence of biopsy complications in pregnancy is variable, varying from 2% to 6.7% [79], with a possible peak at around 25 gestational weeks, and in most cases, hemorrhagic complications do not require blood transfusions. A systematic review of the studies on kidney biopsies in pregnant patients reported the results of 243 biopsies during pregnancy compared with those of 1236 post-partum biopsies. Only four patients with major bleeding complications were reported; all of them occurred after biopsies were performed after delivery [80]. The adverse events occur more frequently in women with decreased kidney function, elevated blood pressure, or coagulation disorder. Normalization of the blood pressure with antihypertensive agents may reduce the risk of a kidney biopsy, with the exception of patients with advanced kidney failure and sclerotic renal vessels. Pregnant women with thrombocytopenic AKI, antiphospholipid syndrome, and those taking aspirin may have a prolonged bleeding time. In those cases, the preoperative infusion of desmopressin (0.3 micromoles/kg over 30 min) can achieve the normalization of the bleeding time for some hours and allow a kidney biopsy to be performed. However, severe hemorrhagic diathesis represents a formal contraindication to kidney biopsy.

## 7. Conclusions

In conclusion, pregnancy is associated with an increased risk of complications in patients with glomerular diseases, and often the differential diagnosis between the reactivation or onset of the glomerular disease and pregnancy complications, such as pre-eclampsia, may be difficult. A correct diagnosis is crucial for the outcome of both the mother and fetus. In this setting, a kidney biopsy is very helpful to obtain a prompt diagnosis to decide the management of these patients. According to the available literature data and our personal experience, performing a kidney biopsy in patients with urinary abnormalities or alteration of the renal function before planning a pregnancy can be useful to establish the ideal timing for conception. Pre-conception counselling and a planning pregnancy in patients with a quiescent disease, stable kidney function, and normal blood pressure may prevent complications. During pregnancy, we suggest that clinicians consider a kidney biopsy possibly before 25 weeks of gestational age to reduce the risk of complications in patients with active urinary sediment, significant proteinuria (>500 mg/day), or a reduction of GFR, in whom less invasive diagnostic tests have failed to elucidate the disease etiology. In addition, we recommend post-partum renal check-ups, including a kidney biopsy, in patients with alterations of the renal function or persistent proteinuria after delivery (Table 1).

## Figures and Tables

**Table 1 jcm-12-01834-t001:** Recommendations for kidney biopsy in pregnant women with glomerular disease. GFR, glomerular filtration rate; LN, lupus nephritis.

**Kidney biopsy before pregnancy**
Perform a kidney biopsy before planning pregnancy in patients with undiagnosed proteinuria, hematuria, or decline in kidney function: - To establish the diagnosis and the best therapeutic strategy before pregnancy; - To decide the ideal time of conception during the stable remission of the underlining disease to reduce the risk of pregnancy complications; - To predict the outcome of pregnancy and to minimize the maternal and fetal risks.
**Kidney biopsy during pregnancy**
Consider a kidney biopsy in pregnant patients with a sudden decrease in GFR or in presence of active urinary sediment or proteinuria >500 mg/day, in whom less invasive diagnostic tests have failed to elucidate disease etiology. - A kidney biopsy during pregnancy is crucial to distinguish glomerular disease first presenting during pregnancy, reactivation of the underlying disease (seen more frequently in patients with LN), or pre-eclampsia; - Minimize the bleeding risks by monitoring coagulation factors and platelet count (especially in case of severe thrombocytopenia) by reducing blood pressure with antihypertensives and by withdrawing aspirin for at least one week before the procedure; - Avoid a biopsy beyond 28 weeks of gestation to minimize the risks associated with the procedure vs. the risk of preterm delivery.
**Post-partum kidney biopsy**
Consider a kidney biopsy in patients that suffered a kidney injury during pregnancy and particularly in those with persistent or increasing proteinuria after delivery to improve long-term health outcomes. - Patients with LN nephritis may experience an increased incidence of renal flares within the 3 months after delivery that may require a kidney biopsy for correct diagnosis and management.

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
