# Peer review of "Kidney Biopsy in Pregnant Women with Glomerular Diseases: Focus on Lupus Nephritis"

_jcm, 2023, doi:10.3390/jcm12051834_

Round 1

Reviewer 1 Report

„Kidney Biopsy in pregnant women with glomerular diseases: focus on lupus nephritis”

This review had the aim to discuss the role of kidney biopsy in the diagnosis, management and establishment of the prognosis of this subcategory of patients with glomerular disease before or during pregnancy and also in their post-partum period, highlighting also the lupus nephritis patients and also the possible complications associated with the procedure.

I consider that the review is relevant for the field, clearly presented in a well-structured manner. The table with the summarisation of the recommendations for kidney biopsy in pregnant women with glomerular disease is well structured and easy to follow.The conclusions are coherent.

The following are the minor issues that I suggest regarding this research:

1.       Reformulate the following phrase from the introduction as is not clear „However, the risk for maternal and fetal complications is increased renal insufficiency, severe hypertension, proteinuria, lupus nephritis, or type of glomerular disease.” –line 35

I declare that I have no conflicts of interests.

Author Response

Thank you for your observation.

We have modifed the phrase as follow 'However, the risk for maternal and fetal complications is increased particularly in patients with renal insufficiency, severe hypertension, proteinuria, lupus nephritis, or type of glomerular disease'. 

Reviewer 2 Report

Morini and coauthors describe indications for kidney biopsy during pregnancy. It is a well-written review that provides the summary of the current literature. 

One would suggest to provide more detailed description and classification of the pathologic findings in kidney biopsies, this will help practicing nephrologists to understand the importance of the data that could be received from the pathology.   

Author Response

Thank you for your suggestion.

We have provided a description of kidney biopsy in particular of pre-eclampsia and differential diagnosis between pre-eclampsia and other glomerulonephritis.